# Efficient In-Context Visual Learning with Trident Block and Cross Blocks

## Abstract

Visual prompt-based large vision models exhibit remarkable performance in a range of vision tasks. However, visual prompting large vision models are computationally intensive and resource-demanding due to their large parameter sizes and the complexity of processing visual prompts, resulting in inefficiencies in speed and memory usage. To tackle these challenges, we propose the Efficient Painter model, which leverages a novel context-aggregated attention based trident block to alleviate cross-task gaps and reduce memory and computation overhead. Furthermore, we introduce a cross-blocks feature union module to capture global contextual information at different levels and speed up training. This architecture mitigates training costs and memory requirements during inference. Our model strikes a balance between speed and memory efficiency, achieving a $19\times$ reduction in floating point operations per second (FLOPs). Moreover, our model is $9\times$ smaller in model size and runs $4.1\times$ and $27\times$ faster during training and inference, respectively. Comprehensive experiments demonstrate that our design effectively processes additional visual prompts and outperforms baseline methods on standard benchmarks like *SIDD* and *LoL* in zero-shot settings, improving performance by 0.4% and 1.2% respectively.

## 1 Introduction

Visual prompt-based Large Vision Models (LVMs) have recently emerged as a powerful approach for various vision tasks without relying on language instructions. Unlike traditional models like CLIP Radford et al. (2021) and Flamingo Alayrac et al. (2022) that depend on language guidance, visual prompt-based LVMs Wang et al. (2023a;b; 2024) operate solely on continuous visual inputs. These models reduce quantization errors from discretization and enable effective in-context visual learning through masked image modeling techniques by aligning the output space to be as continuous as the input images.

Standard LVMs typically utilize encoder-decoder architectures, which are computationally intensive and resource-demanding due to their complex designs. This issue is magnified in visual prompt-based LVMs, as visual prompt-based LVMs must handle additional visual prompt inputs, increasing computational complexity and resource consumption. During inference, processing high dimensional vision prompts leads to higher latency, negatively impacting performance in real-time applications. Conventional methods like quantization and pruning Liu et al. (2021); Mao et al. (2023); Molchanov et al. (2019) aim to reduce computational load but often fail to improve inference time significantly. Recent studies Kao et al. (2022); Mehta & Rastegari (2022b) have identified redundant parameters and frequent memory access operations in multi-head self-attention (MHSA) as bottlenecks, however, they result in reduced accuracy and minimal speed improvement. While some work Pan et al. (2022; 2023); Wu et al. (2022) focus on redesigning transformer blocks to enhance performance, they do not simultaneously address memory and computational efficiency or reduce parameter redundancy. Therefore, these methods not only fail to achieve overall efficiency, but the distinctive manner in which visual prompt-based LVMs incorporate visual tokens into the models also makes them directly inapplicable. Applying these methods would impair the model's in-context visual learning capabilities. Consequently, such barriers complicate the application of classical architectural optimizations.

Building upon the issues outlined above, the primary challenge lies in achieving a delicate balance between overall model efficiency in computation and memory and the in-context learning (ICL)

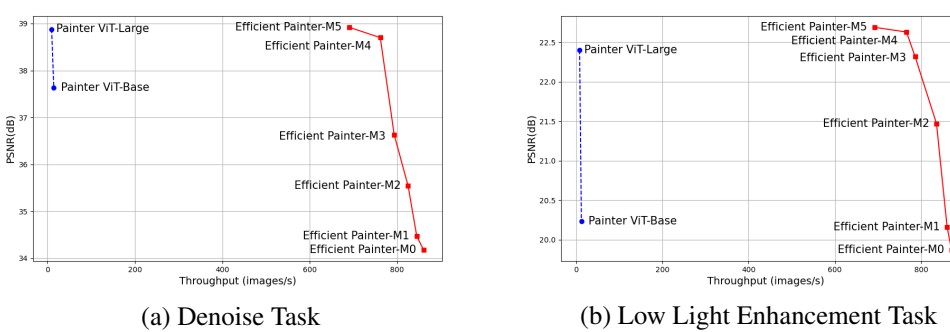

(a) Denoise Task  (b) Low Light Enhancement Task

Figure 1: **The Comparative Analysis Result.** Speed and accuracy compared on 2 benchmarks: *SIDD* and *LoL*. Throughput results are obtained on single GPU A100 (80G).

capacities required for visual prompt-based models. To address this challenge, we propose Efficient Painter, an advanced visual prompt-based LVM. Efficient Painter enhances efficiency by integrating a trident block architecture equipped with context-aggregated attention (CAA), effectively optimizing computational and memory demands while preserving robust in-context visual learning capabilities. This design allows the model to perceive dynamic context information while reducing memory and computational demands. The trident block also decouples visual prompts from task-specific content images, minimizing visual semantic confusion. To enhance the model learning capacity, we integrate a cross-blocks feature union (CBFU) module, which improves multi-level global information processing. Additionally, we optimize prompt embedding orders based on the spatial correlation of inputs, accelerating training and boosting performance. Our model demonstrates substantial improvements in image low-light enhancement and denoising tasks, surpassing previous state-of-the-art (SOTA) models. Besides, as shown in figure 1, we achieved a significant increase in optimized efficiency compared to traditional Painter vision transformer (ViT) variants. The main contributions of this paper are summarized as follows:

- We present a comprehensive study on a popular visual prompt-based LVM Wang et al. (2023a) to identify its main bottlenecks: the additional computational overhead from visual prompts during training and a memory-inefficient architecture design.

- We propose a model named Efficient Painter, which trains the model using an efficient ICL fashion and introduces a CAA based trident block and a CBFU module to alleviate the cross-task gap and reduce memory and computation overhead of the model.

- We perform extensive experiments showing that our design demonstrates superior performance over the baseline methods on standard benchmarks like *SIDD* Abdelhamed et al. (2018) and *LoL* Wei et al. (2018), achieving a speed up of 0.4% and 1.2% respectively.

## 2 RELATED WORK

**Efficient ViTs.** ViTs Wang et al. (2021a) exhibit powerful capabilities in various vision tasks. However, ViTs are hindered from deployment due to the limited throughput in resource-limited environments. Traditional approaches such as DeiT Touvron et al. (2021), and MobileViT Mehta & Rastegari (2022a) leverage knowledge distillation or architectural downsize fail to enhance actual inference speed or throughput. Conversely, post-training techniques like Token Merging (ToMe) Bolya et al. (2023) and DiffRate Chen et al. (2023) optimize token usage to reduce the model's FLOPs. Despite lowering theoretical computational complexity, they do not improve memory efficiency significantly due to additional parameters introduced in Multi-head Self-attention and Feed-Forward Layers. Recent strategies like EfficientSAM Zhao et al. (2023) aim to balance parameter reduction and computation demands by transferring the capabilities of heavy-weight models to smaller ones using masked image pretraining. However, it incurs significant computational and memory overhead during training, making it intolerable in scenarios with limited resources and in vision tasks spanned across multiple semantic contexts. In addition, these methods focus primarily on specific tasks such as SAM, which limits their general applicability.

**LVMs with ICL.** ICL Brown et al. (2020) enables models to adapt to new tasks by leveraging contextual information during inference. Recent research has demonstrated that LVMs are particularly effective for ICL due to their self-attention mechanisms, which are adept at modeling long-range dependencies across images and texts. Prominent ICL ViTs, such as Perceiver IO Jaegle et al. (2021) and Flamingo Alayrac et al. (2022), excel in few-shot learning by effectively utilizing contextual examples. Recent innovations, including Visual Prompting Bahng et al. (2022) and PromptGIP Liu et al. (2024), leverage visual prompts for task guidance. However, these methods typically require larger parameter sizes and substantial GPU memory, limiting their practicality for real-life applications. In this paper, we explore the development of a visual prompt-based ViT designed to improve both time and memory efficiency while achieving ICL capabilities. Our approach aims to address existing challenges and enhance the applicability of visual-prompt ICL models in resource-limited scenarios.

## 3 BACKGROUND & MOTIVATION

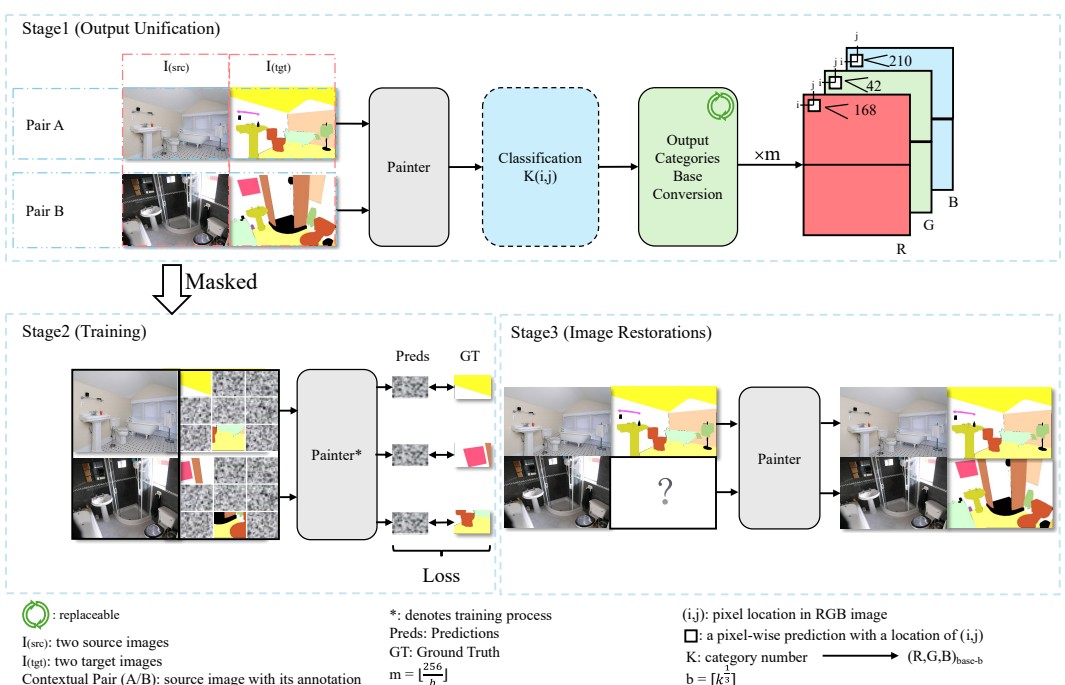

Figure 2: **Training Process of Efficient Painter for Semantic Segmentation. Stage 1:** Semantic categories are converted into three-digit numbers in different bases for pixel-wise representation. For example, "bed" (149 in base 10) becomes 415 in base 6, and when multiplied by margin $m = 42$, results in RGB vector $[168, 42, 210]$. **Stage 2:** Targets are restored using simMIM. **Stage 3:** Predictions are generated using context-related prompts.

**Background.** To achieve in-context visual learning, LVMs like Painter require 3 major stages of the masked image modeling Xie et al. (2022) learning process, as can be shown in Fig. 2. In Stage 1, Painter unifies cross-domain visual tasks into the same RGB image space. During training, as shown in Fig. 2 Stage 2, the Painter utilizes source input pairs: $I_{src}$ as queries and their annotated target counterpart $I_{tgt}$ as answers and applies a 75% mask ratio to the answers. Consequently, during inference as shown in Stage 3. in Fig. 2, the model only requires stacking a task-specific prompt and another source image to complete the pixel-wise reconstruction. The core of the ICL frameworks capable of performing extensive multi-modal vision tasks involves employing ViT-Large backbone, which contains over 307 million parameters Wang et al. (2023a;b); Bar et al. (2022).

**Motivation.** In this chapter, we empirically compare Painter with the SOTA EfficientViT backbone Liu et al. (2023) to assess their overall efficiency. Additionally, leveraging Painter's architecture, we investigate the relationship between efficiency and in-context visual learning capabilities. Table 1

Table 1: **A Heuristic Approach Optimizes Model Size and Reduces Computational Complexity.** For comparative analysis, the dimensions of Query (Q), Key (K), and Value (V) matrices in a Painter-customized ViT-Large block are presented. The Feed-Forward Network (FFN) component within the Efficient Encoder has been reduced by $2\times$ relative to the Painter's Encoder architecture. The embedding dimension (ED) is utilized to define the dimensionality of the ViT blocks. It is noteworthy that the decoder in the Painter architecture is implemented as a series of stacked FFNs.

| Blocks | Heads | Q | K | V | Depth | FFN size | Params(M) | FLOPs(G) |
|---|---|---|---|---|---|---|---|---|
| Decoder (Painter) Wang et al. (2023a) | N/A | N/A | N/A | N/A | N/A | ED $\times 4 \times P^2$ | 67.16 | 119.875 |
| Encoder ViT-Base (Painter) Dosovitskiy et al. (2021) | 12 | 64 | 64 | 64 | 12 | ED $\times 4$ | 87 | 17.58 |
| Encoder ViT-Large (Painter) Dosovitskiy et al. (2021) | 16 | 64 | 64 | 64 | 24 | ED $\times 4$ | 370.7 | 673.24 |
| Efficient-ViT: stage 1 Liu et al. (2023) | 4 | 16 | 16 | 64 | 1 | ED $\times 2$ | 0.359 | 0.074 |
| Efficient-ViT: stage 2 Liu et al. (2023) | 4 | 16 | 16 | 64 | 2 | ED $\times 2$ | 3.551 | 0.209 |
| Efficient-ViT: stage 3 Liu et al. (2023) | 4 | 16 | 16 | 64 | 3 | ED $\times 2$ | 7.959 | 0.141 |

shows that Painter's architecture is inefficient due to heavy encoder blocks. Fine-tuning a ViT-Large backbone on a single A100 80G GPU limits the batch size to 8 and requires approximately 120 GPU hours for near-SOTA performance, yielding a throughput of 8.02 images/s. Using pure visual prompts instead of linguistic guidance expands the input from $224 \times 224$ (196 patches) to $448 \times 448$ (768 patches), leading to quadratic growth in computational and memory demands and significantly increasing the model's FFN embedding dimensions. Replacing the ViT-Large backbone with a ViT-Base variant reduces model complexity but significantly compromises in-context visual learning capabilities without delivering substantial efficiency gains. Specifically, the ViT-Base variant achieves throughput gains of 53.5% and 52% on the *SIDD* and *LoL* datasets, respectively, while resulting in PSNR degradation of 3.22% and 9.69% on these image processing tasks.

These findings suggest that indiscriminate efficiency improvements can compromise critical aspects like in-context visual learning.

## 4 OUR DESIGN

### 4.1 OVERVIEW

Designing an Efficient LVM capable of ICL from visual prompts across diverse visual tasks presents significant challenges due to the varying context information required for different visual tasks. While substituting a more efficient backbone seems like a straightforward approach, transferability will diminish during training.

To overcome such deficiencies, our core design is built on three key components: (a) Trident block with CAA; (b) CBFU with context-based order optimization; (c) context-aware multiscale reconstruction loss function. The overall architecture is depicted in Fig. 3.

### 4.2 EFFICIENT LVMS WITH TRIDENT BLOCK

The following sections first introduce the CAA module and the trident block. As illustrated in Fig. 3 (a), the context-aggregated attention module and the trident block are designed to effectively capture and process in-context information from image pairs while maintaining computational and memory efficiency.

**Context-Aggregated Attention**. Previous studies Wang et al. (2023a); Liu et al. (2024) utilizes computation-bounded MHSA for learning contextual representations. As the contextual information within each image pair for a specific task is spatially correlated and independent of channel information. We partition the input features in a manner that aligns the number of channels with the number of attention heads to enhance the versatility of MHSA and reduce its computational cost, as illustrated in the left part in Fig. 3 (a). Furthermore, during the self-attention process, the channel-wise features are aggregated again by applying an extra convolution layer iteratively to pass global information. Let $F$ denotes the entire contextual feature map serving as the input of the CAA layer, $\Phi$ denotes convolution, and $h$ denotes a smaller number of attention heads. The iterative process can be formalized as follows:

$$F'_j = F_j + \Phi(\tilde{F}_{(j-1)}), \quad 1 < j \leq h$$

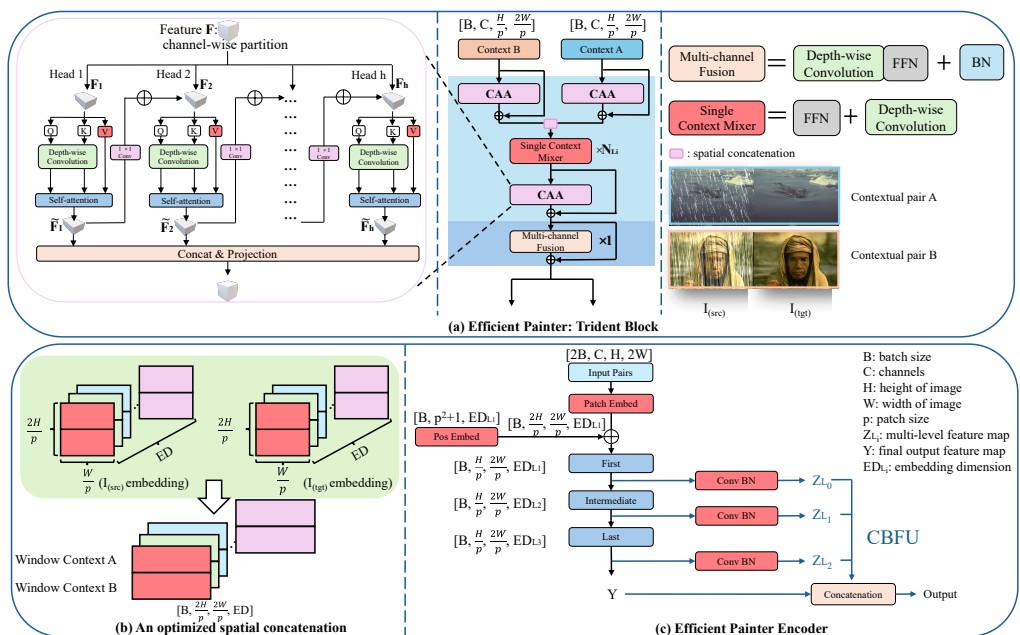

Figure 3: **An Overview of Proposed Efficient Painter Architecture.** (a) The image features three key components: the CAA module at the left, the trident block with two context inputs in the center, and the detailed structure of each component within the trident block on the right. (b) The diagram also includes context-related order optimization. (c) The Efficient Painter encoder is depicted, showcasing variations such as CBFU, aggregation of blocks First, aggregation of blocks Intermediate, and aggregation of blocks Last, each containing a different number of Efficient Painter blocks in the Efficient Painter architecture.

where $F'_j$ is the $j$-th partition of an entire input feature map $F = [F_1, F_2, ..., F_h]$ corresponding to the $j$-th head, and $\tilde{F}_{(j-1)}$ denotes the feature from the previous head. Additionally, CAA modifies the traditional attention mechanism to reduce redundancy in parameters and computational overhead by adopting smaller dimensions for Q and K. Finally, CAA improves contextual locality between Q and K through the depth-wise convolution layer (DWConv) Howard et al. (2017), compared to recent work Liu et al. (2023).

**Trident Block.** Our lightweight encoder is constructed by stacking three levels of aggregation of blocks as shown in the middle part of Fig. 3 (a). In each level of block aggregation, we employ a distinct number of **trident blocks** designed to handle the contextual unrelated visual embeddings in parallel. In particular, two CAA layers, as mentioned in the previous section, are applied to generate representations for two contextual image patches, $F_A$ and $F_B$, shaped as $[B, C, \frac{H}{p}, \frac{2W}{p}]$ from the same task. These patches are subsequently transformed into two task-specific visual tokens with shape of $[B, ED_{L_i}, L_{token}]$, where $L_{token} = \frac{H}{p} \times \frac{2W}{p}$ and $L_i$ denotes the embedding dimensions at the $i$-th level. However, since the generated visual tokens possess only independent contexts, this leads to a reduction in mutual information and results in limited representation abilities. We introduce a **single context mixer** that features a DWConv Howard et al. (2017) layer and an FFN, where the FFN is composed of 2 inverted DWConv Howard et al. (2017), to address this issue.

Despite the current design's efficiency, it needs to work on utilizing the global relationship effectively when the task-specific prompt changes in LVMs. Trident block further incorporates a CAA layer and a **multiple-channel fusion** layer for enhanced context awareness. This multiple-channel fusion layer comprises $N_{L_i}$ DWConv Howard et al. (2017) and FFN based on tailored to its hierarchical level. Additionally, it includes a single Batch Normalization layer, where $N_{L_i}$ represents the number of layers, $L_i$ indicates the corresponding hierarchical level $i$. To maintain the input-output symmetry, we apply another context-wise concatenation before the mixer and a context-wise split after the fusion.

Finally, our trident block can be reformulated as

$$\hat{F}_A, \hat{F}_B = \Phi_{\text{fusion}}(\text{CAA}\{\Phi_{\text{mix}}(\text{CAA}(F_A), \text{CAA}(F_B))\}),$$

where $\Phi_{\text{mix}}$ and $\Phi_{\text{fusion}}$ refer to single context mixer and multiple channel fusion layer. By incrementally integrating more combinations of DWConv Howard et al. (2017) and FFN as the number of trident blocks increases, this arrangement as shown in Table 2, not only minimizes memory consumption but also expedites understanding of the high-dimensional relationship compared to its ViT counterpart. Despite these improvements, significant redundancy in global contextual information remains in the lower tiers of the traditional backbone. To address this issue, we progressively increase the embedding dimensions $ED_{L_i}$ according to hierarchical levels, as illustrated in Table 2. Our trident blocks between each level are designed to reshape the context from $[B, ED_{L_i}, L_{token}]$ into higher dimensional feature maps with a shape of $[B, ED_{L_{i+1}}, L_{token}]$. This transformation ensures effective representation with minimal memory overhead introduced. The trident block depicted in Fig.3 (a) effectively grasps local and global contextual information through paralleled and memory-efficient design, improving efficiency during the training and inference phases.

Table 2: **Architecture Design Details of Efficient Painter Encoder Variants.** We employ a hierarchical structure comprising three levels of trident block aggregation. Each level incorporates $L_i$ trident blocks. Within each level, the trident blocks utilize an equal embedding dimension of $ED_{L_i}$ and $N_{L_i}$ combinations of DWConv Howard et al. (2017) and FFN.

| Model | $\{L_1, L_2, L_3\}$ | $\{N_{L_1}, N_{L_2}, N_{L_3}\}$ | $\{ED_{L_1}, ED_{L_2}, ED_{L_3}\}$ | Q,K dimensions |
|---|---|---|---|---|
| Efficient Painter-M0 | $\{1, 2, 3\}$ | $\{1, 2, 3\}$ | $\{64, 128, 192\}$ | 16 |
| Efficient Painter-M1 | $\{1, 2, 3\}$ | $\{1, 2, 3\}$ | $\{128, 144, 192\}$ | 16 |
| Efficient Painter-M2 | $\{1, 2, 3\}$ | $\{1, 2, 3\}$ | $\{128, 192, 224\}$ | 16 |
| Efficient Painter-M3 | $\{1, 2, 3\}$ | $\{1, 2, 3\}$ | $\{128, 240, 320\}$ | 16 |
| Efficient Painter-M4 | $\{1, 2, 3\}$ | $\{1, 2, 3\}$ | $\{128, 256, 392\}$ | 16 |
| Efficient Painter-M5 | $\{1, 3, 4\}$ | $\{1, 2, 3\}$ | $\{192, 288, 392\}$ | 16 |

## 4.3 COMBINED SINGLE AND MULTI-BLOCKS FEATURE

To ensure our efficient LVMs are robust across a wide distribution of contextual details in the visual prompts and capture global information accurately in our efficient encoder, we propose a CBFU with contextual order optimization.

**Cross Blocks Feature Union.** Traditional element-wise addition or simple concatenation is inadequate for encoding decoupled information between different blocks. Drawing on shortcut design principles, as seen in ResNet and U-Net He et al. (2015); Ronneberger et al. (2015). To improve representational capacity, we implement CBFU module as outlined in Fig. 3 (c) to leverage different levels of abstraction. CBFU integrates three direct shortcut feature maps from distinct blocks of the Efficient Painter architecture, processed through a shared-weight Convolution and Batch Normalization (ConvBN) layer, denoted as $\Psi$. Subsequently, these maps are concatenated with a raw output, $Y$, from the final aggregation blocks. This fusion process can be formalized as follows:

$$\text{CBFU} = \text{concat}\left[\bigcup_{L_i}(\Psi(Z_{L_i})), Y\right],$$

where $Z_{L_i} \in \mathbb{R}^{(B, \frac{H}{P}, \frac{W}{P}, ED_{L_i})}$ as shown in Fig. 3 (c) represents the output feature map at the $i$-th level, with $ED_{L_i}$ denoting the embedding dimension for level $i \in \{1, 2, 3\}$. Following concatenation, an FFN remaps the feature map to align with the dimensions of the subsequent decoder, similar to that in the previous work Wang et al. (2023a). This approach unifies feature maps across the network, and thereby mitigate the limitation in previous study Liu et al. (2023). It offers two primary advantages: alleviation of gradient explosions during training and information loss inherent in the original Painter ViT block; and (2) the ability to utilize a smaller number of multiple-channel fusion layers as illustrated in trident block.

**Order Optimization.** As illustrated in the training process detailed in the background section, prior work Wang et al. (2023a) constructs a pair of queries and a pair of answers from $I_{(src)}$ and $I_{(tgt)}$ and feed them through patch embedding layer seperately. This approach shows a limited

representational capability for pixel-wise reconstruction tasks where the input and output spaces are highly similar. To address this limitation, we adopt the method depicted in Fig. 3 (b), which employs multi-scale convolutional layers to create patch embedding. Additionally, queries and answers are horizontally concatenated into "window context" embeddings in the spatial domain based on contextual information, thereby enhancing the model's cross-task awareness.

## 4.4 CONTEXT-AWARE RECONSTRUCTION OBJECTIVE FUNCTION

During training, Efficient-Painter consists of a context-reorganizing encoder and a conventional dense pixel-wise predictor as the decoder. Upon observation in Appendix Fig. 5, using smooth-$l_1$, a clear and increasing halo of noise is apparent around the edges of the artifact. Inspired by the approach in Zhao et al. (2017), we adopted a perceptual loss function combining MS-SSIM and $l_1$ to alleviate this problem,

$$\hat{\mathcal{L}} = \alpha \cdot L_{\text{MS-SSIM}}(I_{src}, I_{tgt}) + (1 - \alpha) \cdot l_1(I_{src}, I_{tgt}),$$

the detail of MS-SSIM loss function is included in the Appendix Loss Function A.3. However, we still identify an inefficiency during training by using perceptual loss function. Inspired by that ICL visual inpainting is essentially a prompt-guided pixel-wise reconstruction process, we propose another complementary context-aware loss function that incorporates a pretrained Painter encoder for feature alignment tuning. Denote the final output feature map of a pretrained Painter encoder as $f_p(X_i; \hat{\theta})$, and the counterpart feature map of efficient Painter encoder as $f_e(X_i; \theta)$, where $X_i$ is the $i$-th data pair concatenated from $I_{src}^i$ and $I_{tgt}^i$ for the same task, $\hat{\theta}$ represents frozen parameters, and $\theta$ represent parameters to be trained. Additionally, $\Omega(d)$ signifies the sum of elements in a batch feature $f(\vec{X}; \theta)$, where $\vec{X}$ is a batch of inputs. Hence, our context-aware loss function can be formulated as follows:

$$L = \beta \cdot \hat{\mathcal{L}} + (1 - \beta) \cdot \frac{1}{\Omega(d)} l_2[f_p(\vec{X}; \hat{\theta}), f_e(\vec{X}; \theta)].$$

By incorporating the context-aware-based $l_2 - \text{norm}$, we leverage the pretrained encoder's knowledge to effectively guide the lightweight model without needing heavy architectures and effectively reduce the iterations needed to reach an optimal solution. Our central objective is to perceive the contextual inductive bias in parallel and embody prior knowledge delivered from a heavy encoder. Meanwhile, our model benefits from different network configurations and can be quickly adapted to other low-level image-processing tasks.

## 5 EVALUATION

### 5.1 EXPERIMENT SETUP

**Hardware and Software.** We compare Efficient Painter with the latest generalist model Painter on seven benchmarks. Speed and accuracy tests are carried out on a single 80G Nvidia A100 (with a peak performance of 9.7T FLOPs) for GPU and AMD EPYC 7543 32-Core Processor for CPU at 2.8 GHz. Our models are built with PyTorch 1.14 Paszke et al. (2019) and detectron2 0.6.0. Other training dependencies are listed in Appendix Table 8.

**Implementation Details.** Our efficient painter model undergoes 20 epochs of training using the AdamW Kingma & Ba (2017) optimizer with the cosine learning rate scheduler. We use a batch size of 256 and a standard data augmentation including random resized cropping and color jittering during training and zero data augmentation during validation. The input contextual pairs are set to $448 \times 224$. We employ a learning rate of $2.4 \times 10^{-3}$, weight decay 0.05 with $[\beta_1, \beta_2] = [0.9, 0.999]$. Our dataset of 434,850 data points combines multiple tasks with customized sampling rates: NYUv2 (0.1), ADE-20K (0.2), COCO (0.6), SIDD (0.15), Derain Scenes (0.05), and LoL (0.05). For our context-aware reconstruction loss function, we set $\alpha = 0.84$, following Zhao et al. (2017). Additionally, we assign $\beta = 0.5$ to balance the complementary and perceptual loss functions.

### 5.2 BENCHMARKS AND DATASETS

We compare our methods with SOTA approaches using eight standard metrics (see Metrics in Appendix A.2). For each of the seven tasks, $y_{i,j}^t$ represents the ground truth and $\hat{y}_{i,j}^t$ the prediction.

Table 3: **Computation Efficiency Comparison.** Comparison of Efficient Painter models and Painter in terms of computation efficiency metrics.

| Model | Epochs | FLOPs (M) | Training Cost (GPU Hours/Epoch) | Avg. Throughput (imgs/s) | Global Degradation (%) |
|---|---|---|---|---|---|
| Efficient Painter-M0 | 20 | 4999 | 2.66 | 877.82 | 6.37 |
| Efficient Painter-M1 | 20 | 5639 | 2.75 | 862.37 | 4.65 |
| Efficient Painter-M2 | 20 | 6010 | 2.89 | 835.84 | 2.58 |
| Efficient Painter-M3 | 20 | 6917 | 3.04 | 786.05 | 1.69 |
| Efficient Painter-M4 | 20 | 7496 | 3.16 | 766.23 | 1.44 |
| Efficient Painter-M5 | 20 | 8957 | 3.32 | 691.75 | 1.36 |
| Painter (ViT-Large) | 15 | 172891 | 13.25 | 8.02 | - |

For instance, monocular depth is evaluated using the root mean square error (RMSE), absolute relative error (A.Rel), and threshold accuracy $\delta_1$ within a range of $[0, 10]$ meters. We utilize a diverse set of datasets for our experiments: For depth estimation, we employ *NYUv2* Nathan Silberman & Fergus (2012). Semantic segmentation leverages *ADE20K* Nathan Silberman & Fergus (2012) (36K training, 650 validation images), while panoptic segmentation uses *MS-COCO* Lin et al. (2015) (110K training, 5K validation images). Keypoint detection utilizes COCO's subset Xiao et al. (2018a) of over 15K annotated samples. Image denoising employs *SIDD* Abdelhamed et al. (2018), comprising 96K noisy images from 10 scenes. For low-light enhancement, we use *LoL* Wei et al. (2018) (500 image pairs: 485 training, 15 testing). Lastly, the *derain* task combines five major datasets Fu et al. (2017), totaling approximately 13K images for training and validation.

## 5.3 MAIN RESULTS

**Computational Efficiency.** In Table 3, we quantitatively evaluate the computational cost of our architecture. Without significant degradation, our Efficient Painter variants achieve a $4.98\times$ GPU-hours training speedup per epoch when using the same dataset. They also provide actual speedup during inference with relatively small FLOPs. Extensive experiments comparing various Efficient Painter model variants (M0 to M5) with the standard Painter (ViT-Large) model demonstrate that by adjusting the number of blocks and embedding dimensions ($ED_{L_i}$), we can optimize the models for improved performance with fewer parameters.

**Memory Efficiency.** As shown in Table 4, increasing complexity from Efficient Painter-M0 to M5 raises parameters from 8.1 million to 39.0 million while decreasing global degradation from 6.37% to 1.36%. For instance, Efficient Painter-M0, with 6 blocks and embedding dimensions $\{64, 128, 192\}$, exhibits a degradation of 6.37%, whereas Efficient Painter-M5, with 7 blocks and dimensions $\{192, 288, 392\}$, reduces degradation to 1.36%. Compared to Painter (ViT-Large), which has 370.7 million parameters and 24 blocks, making it 9.5 times larger than our largest model, Efficient Painter models achieve lower performance degradation with significantly fewer parameters. Even with only 1/19-th the FLOPs, Efficient Painter-M5 matches Painter ViT-Large in accuracy, confirming the efficiency of our design.

Table 4: **Memory Efficiency Comparison.** Comparison of Efficient Painter model variants and Painter in terms of parameters that impact memory efficiency. Global degradation is aggregated across tasks.

| Model | Input size | # of Blocks | $ED_{L_i}$ | Params (M) | Global Degradation (%) |
|---|---|---|---|---|---|
| Efficient Painter-M0 | $448 \times 224$ | 6 | $\{64, 128, 192\}$ | 8.1 | 6.37 |
| Efficient Painter-M1 | $448 \times 224$ | 6 | $\{128, 144, 192\}$ | 11.7 | 4.65 |
| Efficient Painter-M2 | $448 \times 224$ | 6 | $\{128, 192, 224\}$ | 15.1 | 2.58 |
| Efficient Painter-M3 | $448 \times 224$ | 6 | $\{128, 240, 192\}$ | 23.4 | 1.69 |
| Efficient Painter-M4 | $448 \times 224$ | 6 | $\{128, 256, 392\}$ | 29.0 | 1.44 |
| Efficient Painter-M5 | $448 \times 224$ | 7 | $\{192, 288, 392\}$ | 39.0 | 1.36 |
| Painter (ViT-Large) | $448 \times 224$ | 24 | 1024 | 370.7 | - |

**Results**. As shown in table 5, our comprehensive training and evaluation across seven different tasks, including runtime benchmarks, confirm that Efficient Painter achieves a $5.5\times$ speedup with comparable accuracy after 20 epochs. Performance comparisons reveal that Efficient Painter significantly outperforms its counterparts in various benchmarks, demonstrating only minor accuracy drops in tasks like semantic segmentation and panoptic segmentation, while achieving improvements in low-level tasks like PSNR and SSIM. These results highlight Efficient Painter's advantages in

Table 5: **Comparative Analysis of LVMs and Specialized Models Across a Spectrum of Visual Tasks.** we compare with the best results of each method. The backbones of the listed generalist methods are: ViT-Large for UViM, Unified-IO$_{XL}$ with 2925M parameters, ViT-Base with another Transformer decoder for Pix2Seq v2, and ViT-Large for Painter. The N/A indicates that acceleration methods like Efficient-ViT are not applicable to visual prompt-based LVMs.

| | depth estimation NYUv2 | | semantic seg. ADE-20K | panoptic seg. COCO | keypoint det. COCO | denoising SIDD | | deraining 5 datasets | | enhance. LoL | |
|---|---|---|---|---|---|---|---|---|---|---|---|
| | RMSE/A.Rel. | $\delta_1\uparrow$ | mIoU↑ | PQ↑ | AP↑ | PSNR↑ | SSIM↑ | PSNR↑ | SSIM↑ | PSNR↑ | SSIM↑ |
| specialized models | | | | | | | | | | | |
| DenseDepth Alhashim & Wonka (2019) | 0.465/0.123 | 0.846 | - | - | - | - | - | - | - | - | - |
| BinsFormer Li et al. (2022) | 0.330/0.094 | 0.925 | - | - | - | - | - | - | - | - | - |
| UperNet-ViT-Large Xiao et al. (2018b) | - | - | 49.9 | - | - | - | - | - | - | - | - |
| Mask2Former Cheng et al. (2022) | - | - | 57.7 | 57.8 | - | - | - | - | - | - | - |
| DETR Carion et al. (2020) | - | - | - | 45.6 | - | - | - | - | - | - | - |
| HRNet Wang et al. (2020) | - | - | - | - | 76.3 | - | - | - | - | - | - |
| HRFormer Yuan et al. (2021) | - | - | - | - | 77.2 | - | - | - | - | - | - |
| Uformer Wang et al. (2021b) | - | - | - | - | - | 39.89 | 0.960 | - | - | - | - |
| MPRNet Mehri et al. (2020) | - | - | - | - | - | 39.71 | 0.958 | 32.73 | 0.921 | - | - |
| MIRNet-v2 Zamir et al. (2020) | - | - | - | - | - | 39.84 | 0.959 | - | - | 24.74 | 0.851 |
| generalist framework, specialized models | | | | | | | | | | | |
| UViM Kolesnikov et al. (2022) | 0.467/- | - | - | 45.8 | - | - | - | - | - | - | - |
| generalist models | | | | | | | | | | | |
| Unified-IO Lu et al. (2022) | 0.385/- | - | - | - | - | - | - | - | - | - | - |
| Pix2Seq v2 Chen et al. (2022) | - | - | - | - | 64.8 | - | - | - | - | - | - |
| PromptGIP Liu et al. (2024) | - | - | - | - | - | - | - | 25.46 | 0.8399 | 20.30 | 0.803 |
| Painter Wang et al. (2023a) | 0.288/0.080 | 0.950 | 49.9 | 43.4 | 72.1 | 38.88 | 0.954 | 29.49 | 0.868 | 22.40 | 0.872 |
| Painter (Efficient-ViT-M5) | N/A | N/A | N/A | N/A | N/A | N/A | N/A | N/A | N/A | N/A | N/A |
| Efficient Painter (ours) | 0.284/0.076 | 0.94 | 49.5 | 43.2 | 70.7 | 38.92 | 0.960 | 29.61 | 0.842 | 22.69 | 0.884 |

resource consumption, processing speed and accuracy, making it an efficient and versatile solution for real-world applications where both performance and efficiency are critical.

## 5.4 ABLATION STUDY

To gain deeper insights into the factors that influence the speed-accuracy tradeoff in our design, we thoroughly analyze the impact of each component in Efficient Painter. The evaluations are conducted on the low-light enhancement task (LoL Wei et al. (2018)), using our proposed Efficient Painter-M5 model. Each component's modification and its corresponding effect on both performance metrics and computational efficiency are assessed to better understand their contributions to the overall architecture. The results are presented in Table 6.

Table 6: **The Effects of Our Design.** Efficient Painter-M5 on LoL task Wei et al. (2018) with PSNR, SSIM and GPU throughput.

| # | Ablation | Throughput (imgs/s) | Performance | |
|---|---|---|---|---|
| | | | PSNR↑ | SSIM↑ |
| 1 | Efficient Painter-M5 | 691.8 | 22.69 | 0.88 |
| 2 | CAA → MHSA | 224.7 | 21.45 | 0.82 |
| 3 | Number of trident blocks at intermediate level = 2→1 | 703.4 | 20.32 | 0.79 |
| 4 | CBFU → None | 696.8 | 18.49 | 0.64 |
| 5 | Context-aware loss → smooth-$l_1$ | - | 20.24 | 0.73 |

**CAA.** Changing the attention mechanism from CAA to MHSA drastically reduced throughput to 224.7 images per second and decreased the PSNR to 21.45 and SSIM to 0.82. This suggests that while MHSA is a powerful attention mechanism, it is computationally more intensive and less effective in this context than CAA.

**Number of trident blocks at intermediate level.** Reducing the number of trident blocks at the intermediate level from 2 to 1 led to a decrease in both PSNR (from 22.69 to 20.32) and SSIM (from 0.88 to 0.79). The decline in image quality metrics suggests that fewer intermediate blocks limit the model's ability to capture and refine the key features required for effective image enhancement.

**CBFU.** Eliminating the CBFU entirely resulted in a substantial drop in performance, with PSNR dropping from 22.69 to 18.49 and SSIM declining from 0.88 to 0.64. Throughput remained relatively stable, this indicates that the feature union plays a critical role in the model's ability to enhance images effectively.

**Contextual Order Optimization.** Switching from context-aware loss to smooth-$l_1$ loss resulted in a moderate reduction in performance, with PSNR dropping to 20.24 and SSIM to 0.73.

## 5.5 DISCUSSION

Table 7: **Comparative Computational Complexity of CAA vs. MHSA.** For a fair comparison, we use the same embedding dimension $ED$ in both the CAA of our efficient trident block and the MHSA in the ICL ViT-Large. For simplicity, we assume that: embedded patch token $x_p$ with a shape of $[B, C, H_p, W_p]$, which will be embeded into $[B, ED, H_p \times W_p]$ in the attention module, we compare their $\mathcal{O}(\cdot)$ complexities based on the main computational overheads. Here, $n = H_p \cdot W_p$, and $h$ is the number of heads

| Computation overhead in Attention | MHSA ($\mathcal{O}(n)$) | CAA ($\mathcal{O}(n)$) | Speedups $\times$ |
|---|---|---|---|
| Linear Embedding | $B \times n \times 3 \cdot ED^2$ | $B \times n \times 3 \cdot \left(\frac{ED}{2}\right)^2$ | $h^2$ |
| Attention Score ($Q \cdot K^T$) | $B \times n^2 \times ED$ | $B \times n^2 \times \frac{ED}{4h}$ | 4h |
| Applying Softmax | $B \times n^2$ | $B \times n^2$ | - |
| Output Attention | $B \times n^2 \times \frac{ED}{h}$ | $B \times n^2 \times \frac{ED}{4h}$ | 4 |

**Computational Efficiency in CAA.** As shown in Table 7, we provide a detailed analysis of the CAA's computation efficiency with MHSA. Specifically, setting the dimensions of Q and K to $\frac{1}{4}$ to that of V in CAA, our approach reduces the complexity of the calculation by at least $4\times$ in each operations. Furthermore, we also identifies that employing the CAA method, which separates contextual features, enables effective capture of contextual information even when utilizing the same masking strategy as shown in Fig. 4.

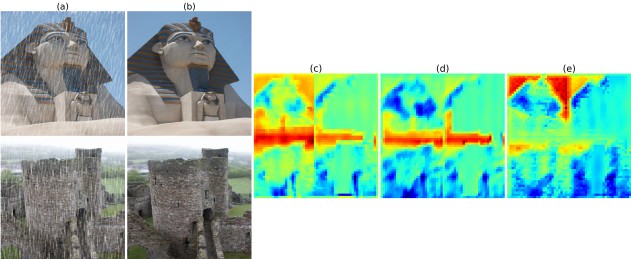

Figure 4: **Comparison of CAA vs. MHSA.** (a) rainy image pairs (b) ground truth pairs (c) non-feature splits, where MHSA is applied (d) CAA with 8 channel splits (e) CAA with 16 channel splits. To enhance visualization, we extract the same level of global average feature map after only applying 50% consistent mask to all input target pairs.

## 6 CONCLUSION

Large Vision Models with ICL ability that employ pure vision guidance demonstrate great potential for solving cross-task visual tasks. In this paper, we introduce Efficient Painter, an architecture designed to optimize the speed-accuracy trade-off of a popular LVM: Painter. By leveraging an improved trident block with CAA and a new CBFU module, Efficient Painter significantly reduces training GPU hours to one-third and increases the throughput by 27 to 44 times, while maintaining a comparable precision. In addition, Efficient Painter surpasses some current vision generalist models in low-level tasks, providing a viable solution for deploying LVMs on resource-constrained devices.

## 7 ETHIC STATEMENT

The authors declare that there are no conflicts of interest with any individuals at Purdue University during the review process.

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

# A APPENDIX

## A.1 MORE RESULTS

## A.2 METRICS

- average relative error (A. Rel.):

$$\frac{1}{n} \sum_{i,j} \left| \frac{y_{i,j}^t - \hat{y}_{i,j}^t}{y_{i,j}^t} \right|;$$

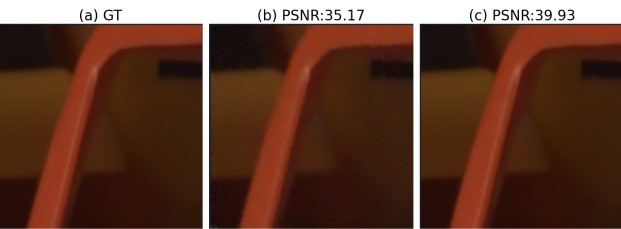

Figure 5: **Visualization of Different Loss Function on Denoise Task.** (a) Groundtruth (b) Smooth-$l_1$ loss function is adopted. (c) Context-aware loss function is adopted. It alleviates image degradations, such as blurring, and produces higher-quality images during reconstruction..

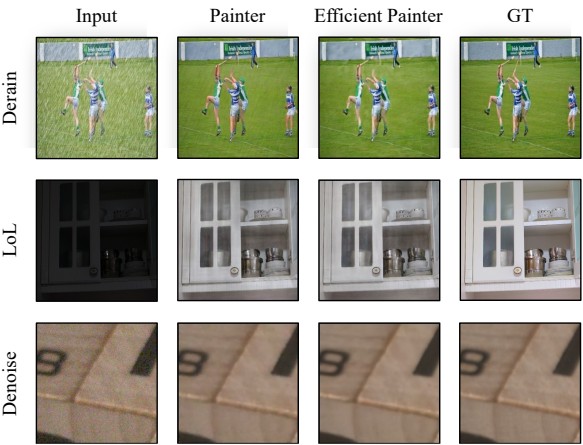

Figure 6: **Comparative Visualization.** Our model outperforms Painter on 3 low-level reconstruction tasks. Input and groundtruth (GT) are presented as well.

- root mean squared error (RMSE):

$$\sqrt{\frac{1}{n}\sum_t (y_{i,j}^t - \hat{y_{i,j}^t})^2};$$

- average ($\log_{10}$) error:

$$\frac{1}{n}\sum_t \left|\log_{10}(y_{i,j}^t) - \log_{10}(\hat{y}_{i,j}^t)\right|;$$

- threshold accuracy ($\delta_i$): % of $y_{i,j}^t$ s.t. $\max\left(\frac{y_{i,j}^t}{\hat{y}_{i,j}^t}, \frac{\hat{y}_{i,j}^t}{y_{i,j}^t}\right) = \delta < $ thr for thr $= 1.25, 1.25^2, 1.25^3$

- mean Intersection over Union (mIoU):

$$mIoU = \frac{1}{k+1}\sum_{i=0}^{k} \frac{p_{ii}}{\sum_{j=0}^{k} p_{ij} + \sum_{j=0}^{k} p_{ji} - p_{ii}}$$
$$= \frac{TP}{(FP + FN + TP)}$$

- $k$ is the number of classes,
- $p_{ii}$ represents the true positives for class $i$,
- $p_{ij}$ represents the predictions of class $i$ as class $j$,
- $p_{ji}$ represents the predictions of class $j$ as class $i$.

- peak signal to noise ratio (PSNR):

$$\text{PSNR} = 10 \cdot \log_{10}\left(\frac{\text{MAX}_I^2}{\text{MSE}}\right)$$

where $\text{MAX}_I$ is the maximum possible pixel value of the image and MSE is the Mean Squared Error between the original and the desired output image.

- structural similarity index (SSIM):

$$\text{SSIM}(x, y) = \left[l(x,y)^\alpha \cdot c(x,y)^\beta \cdot s(x,y)^\gamma\right] = \frac{(2\mu_x\mu_y + c_1)(2\sigma_{xy} + c_2)}{(\mu_x^2 + \mu_y^2 + c_1)(\sigma_x^2 + \sigma_y^2 + c_2)}$$

where:

  - $l(x,y)$, $c(x,y)$, $s(x,y)$ represent luminance comparison, contrast comparison, and structure comparison respectively;
  - $\mu_x$, $\mu_y$ represent the mean of $x$, $y$ respectively;
  - $\sigma_x^2$, $\sigma_y^2$ represent the variance of $x$, $y$ respectively;
  - $\sigma_{xy}$ is the covariance of $x$ and $y$;
  - $c_1 = (k_1L)^2$, $c_2 = (k_2L)^2$ are two variables to stabilize the division with a weak denominator;
  - $L$ is the dynamic range of the pixel-values (typically this is $2^{bits\ per\ pixel} - 1$);
  - $k_1 = 0.01$ and $k_2 = 0.03$ by default.

- panoptic quality (PQ) for panoptic segmentation:

$$PQ = \underbrace{\frac{\sum_{(p,g)\in TP}\text{IoU}(p,g)}{|TP|}}_{\text{segmentation quality (SQ)}} \times \underbrace{\frac{|TP|}{|TP| + \frac{1}{2}|FP| + \frac{1}{2}|FN|}}_{\text{recognition quality (RQ)}}$$

## A.3 LOSS FUNCTION

The MS-SSIM loss function $L_{\text{MS-SSIM}}$ between two images $\mathbf{x}$ and $\mathbf{y}$ is defined as:

$$L_{\text{MS-SSIM}}(\mathbf{x}, \mathbf{y}) = 1 - \text{MS-SSIM}(\mathbf{x}, \mathbf{y}) \tag{1}$$

The MS-SSIM index is computed as the weighted product of SSIM indices across multiple scales:

$$\text{MS-SSIM}(\mathbf{x}, \mathbf{y}) = \prod_{j=1}^{M}\left[\text{SSIM}_j(\mathbf{x}, \mathbf{y})\right]^{w_j} \tag{2}$$

Each SSIM index at scale $j$ is given by:

$$\text{SSIM}_j(\mathbf{x}, \mathbf{y}) = \frac{(2\mu_{\mathbf{x},j}\mu_{\mathbf{y},j} + C_1)(2\sigma_{\mathbf{xy},j} + C_2)}{(\mu_{\mathbf{x},j}^2 + \mu_{\mathbf{y},j}^2 + C_1)(\sigma_{\mathbf{x},j}^2 + \sigma_{\mathbf{y},j}^2 + C_2)} \tag{3}$$

**Notation details**

- $\mathbf{x}, \mathbf{y}$: Input images between which similarity is to be measured.
- $L_{\text{MS-SSIM}}(\mathbf{x}, \mathbf{y})$: MS-SSIM Loss Function, defined as one minus the MS-SSIM index.
- $\text{MS-SSIM}(\mathbf{x}, \mathbf{y})$: Multi-Scale Structural Similarity Index, representing the overall similarity between images $\mathbf{x}$ and $\mathbf{y}$ across multiple scales.
- $M$: Total number of scales at which SSIM is computed. Increasing $M$ allows capturing structural information at more levels of detail.
- $\text{SSIM}_j(\mathbf{x}, \mathbf{y})$: Structural Similarity Index at scale $j$. It measures the similarity between $\mathbf{x}$ and $\mathbf{y}$ at a specific scale.
- $w_j$: Weight assigned to the SSIM index at scale $j$. The weights typically satisfy $\sum_{j=1}^{M} w_j = 1$, ensuring a normalized contribution from each scale.
- $\mu_{\mathbf{x},j}, \mu_{\mathbf{y},j}$: Local means of images $\mathbf{x}$ and $\mathbf{y}$ at scale $j$. These are usually computed using a Gaussian filter.
- $\sigma_{\mathbf{x},j}^2, \sigma_{\mathbf{y},j}^2$: Local variances of images $\mathbf{x}$ and $\mathbf{y}$ at scale $j$.
- $\sigma_{\mathbf{xy},j}$: Local covariance between images $\mathbf{x}$ and $\mathbf{y}$ at scale $j$.

## A.4 LOSS TREND PLOT

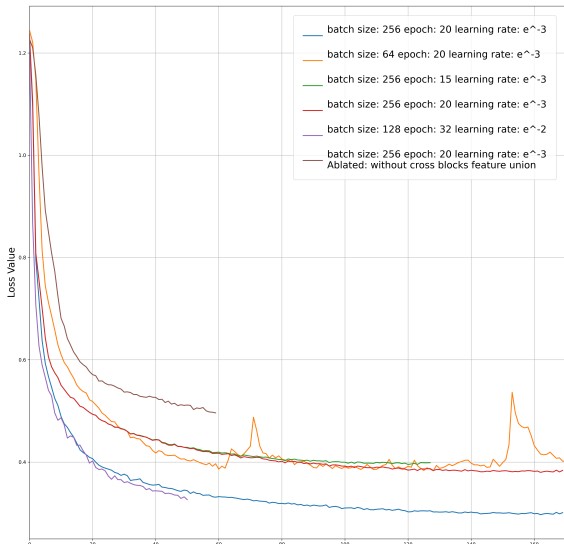

Figure 7: **Loss Trends for Varying Hyper-parameters and Cross-Block Feature Union (CBFU) Ablation.** This graph shows the progression of training loss across different configurations of the Efficient Painter-M5, highlighting the impact of batch size, learning rate, and epoch adjustments, as well as the significant effect of removing the CBFU. Each line represents a distinct setup, providing a visual comparison of how these changes influence model training and efficiency.

## A.5 SOFTWARE ENVIRONMENT

Table 8: Software Environment

| Dependency | Version |
|---|---|
| detectron2 | 0.6.0 |
| fairscale | 0.4.13 |
| fvcore | 0.1.5 |
| h5py | 3.10.0 |
| panopticapi | 0.1 |
| timm | 0.5.4 |
| torch | 0.13.1+cu117 |
| torchvision | 0.14.1+cu117 |
| torchaudio | 0.13.1+cu117 |
| xtcocotools | 1.14.3 |
| yacs | 0.1.8 |

