# OpenReview forum: "Efficient In-Context Visual Learning with Trident Block and Cross Blocks"
_ICLR.cc/2025/Conference — Submitted to ICLR 2025_

### Official Review · Reviewer_JQqw · 2024-10-25

**Soundness:** 3
**Presentation:** 3
**Contribution:** 3
**Rating:** 6
**Confidence:** 4

**Summary:**

This paper mainly focuses on visual in-context learning, specifically the previous work Painter. The authors point out that the previous in-context visual learning methods require unaffordable computational resources. To this end, the authors propose a more efficient network structure specifically designed for visual in-context learning, which can alleviate cross-task gaps and reduce memory and computation overhead.

**Strengths:**

1. The method is clearly explained.
2. The idea of reducing computational complexity for visual in-context learning is interesting and practical.

**Weaknesses:**

1. I am not sure how the proposed CAA is implemented. It seems the information is sequentially propagated between different attention heads.
2. It seems the proposed method may be overly complex. To achieve the goal of better efficiency, the authors rebuild almost all basic modules in the original Painter. While there are several ablation studies showing the efficacy of these designs, these experiments only focus on the high-level design. It is still not clear if it is necessary to build the modules the same as in L266 and L306.
3. I am not sure why the authors have to include the Painter+EViT with all NaN restuls in Tab.5?
4. I wonder if the author can compare the proposed method with methods such as distillation and pruning?
5. It would be better if the authors can test the proposed methods on different in-context learning tasks. For example, use MAE-VQGAN [1] for segmentation, detection and colorization.



[1] Bar A, Gandelsman Y, Darrell T, et al. Visual prompting via image inpainting[J]. Advances in Neural Information Processing Systems, 2022, 35: 25005-25017.

**Questions:**

Please refer to the weaknesses.

---

> ### Author Response · Authors · 2024-11-27
>
> Dear Reviewer JQqw,
>
> Thank you for your thorough review and insightful comments on our manuscript. We appreciate your time and effort in providing detailed feedback, which has been invaluable in helping us improve our work. We have addressed each of your concerns below.
>
> **Weakness W1:**
>
> Thank you for your observation. **Your understanding is correct. In our Context-Aggregated Attention (CAA) module, information is sequentially propagated between different attention heads.** Despite this sequential processing, our comprehensive analysis, as presented in Table 7, demonstrates the superior computational efficiency of CAA compared to traditional Multi-Head Self-Attention (MHSA) under identical embedding dimensions. The efficiency gains are achieved through multiple innovative strategies:
>
> - Quantitative Comparison:** Given an input of intermediate window features ($x_p$) with shape ($[B, ED, H_p, W_p] = [1, 1024, 14, 14]$) and an identical embedding dimension ($ED = 1024$), our benchmarks show:
>
>     - **CAA:** 242.50 MFLOPs
>     - **MHSA:** 822.08 MFLOPs (3.07× higher)
>
> - **Dimension Reduction:** We utilize $\frac{1}{4}$ of the query (Q) and key (K) dimensions compared to traditional MHSA, significantly reducing computation without sacrificing performance.
>
> - **Spatially Localized Convolutional Embeddings:** By replacing dense fully connected embeddings (linear layers) with spatially localized convolutional embeddings, we reduce computational demand, which is particularly beneficial for resource-constrained devices during the training stage.
>
> These combined strategies result in at least a 4-fold improvement in computational efficiency across multiple stages of attention computation. **Therefore, even though the heads in CAA cannot run in parallel due to the sequential propagation, the overall efficiency is significantly improved compared to MHSA.**
>
> **Weakness W2:**
>
> Thank you for your critical observation. **Our goal is to structurally modify visual prompt-based LVMs like Painter, aiming to achieve both high overall efficiency and excellent in-context visual learning capabilities, rather than merely simplifying complexity to achieve overall efficiency.**
>
> **Regarding Line 266:** Our trident block design is pretty straightforward. It aims to represent two distinct contextual visual tokens (patches) in parallel using two independent CAAs. Then, we use 2 inverted parameter-efficient depthwise convolution  (DWConv) layers to mimic the FFN in the original ViT block to fuse these features.
> Finally, this fused information is decoupled to maintain input-output symmetry between trident blocks, to yield context-based embeddings. This design choice not only enhances efficiency but also ensures the input-output symmetry between each block.
>
> **Regarding Line 306:** For our Cross Blocks Feature Union (CBFU), as validated in Table 6, Entry 4, in Section 5.4, **we confirmed the effectiveness of CBFU in maintaining accuracy and its lightweight characteristics. Moreover, it keeps the relative integrity of Painter’s subsequent decoder.**
>
> These module designs are crucial for achieving our objectives, and the ablation studies at both high-level and module-specific levels demonstrate the necessity of these architectural choices. We will provide more detailed explanations and additional experiments in the future to further optimize the basic modules as outlined. But for now, the current results already prove that our design carefully meets our aim.

---

> ### Author Response · Authors · 2024-11-27
>
> **Weakness W3:**
>
> Thank you for highlighting the issue with Table 5. **The inclusion of "NaN/NaN" placeholders for Painter (Efficient-ViT-M5) in Table 5** was intended to indicate that directly applying certain optimization methods designed for single-task models to visual prompt-based models like Painter without careful consideration is not applicable as baseline networks for comparison in our study, also doesn't meet the objective of our paper: to structurally refine a visual prompt-based LVM, aiming to simultaneously achieve both high efficiency and robust in-context visual learning capabilities.
>
> Although these designs effectively enhance model efficiency in specific tasks, they cannot be directly applied to visual prompt-based LVMs. They are mainly designed for single-task scenarios and lack extensibility to visual prompt mechanisms. Even if we forcibly apply these compression methods to Painter, they significantly impact its in-context visual learning capabilities. We demonstrate this by adapting the Painter architecture from ViT-Large to ViT-Base and fine-tuning it on three low-level datasets for comparative analysis—an experiment not conducted in the original Painter study. As shown in Figures 1a and 1b, the ViT-Base configuration did not significantly improve throughput and markedly compromised the model’s in-context visual learning capabilities.
>
> We acknowledge that the presentation of our experimental results needs to be more clear. In the revised version of the paper, we updated Table 5 to change the “NAN” placeholders for Painter (Efficient-ViT-M5) to “N/A” and added an explanation about this line of "N/A" placeholders for Painter (Efficient-ViT-M5).
>
> **Weakness W4:**
>
> Techniques like quantization and pruning (Liu et al., 2021; Mao et al., 2023; Molchanov et al., 2019) may not meet our comparison baseline requirements because as mentioned in the second paragraph of our introduction: **"During inference, processing high dimensional vision prompts leads to higher latency, negatively impacting performance in real-time applications. Conventional methods such as distillation and pruning aim to reduce computational load but often fail to improve inference time significantly."**
>
> **Weakness W5:**
>
> We appreciate your valuable suggestion. **Given our current computational resource limitations, we are considering extending our method to MAE-VQGAN in future work.** This extension will serve to validate the effectiveness of our lightweight design for visual prompt-based LVMs across a broader range of tasks.
>
> Once again, we appreciate your valuable feedback.

---

### Official Review · Reviewer_fA3M · 2024-11-01

**Soundness:** 2
**Presentation:** 3
**Contribution:** 2
**Rating:** 5
**Confidence:** 3

**Summary:**

While Visual prompt-based large vision models exhibit remarkable performance in a range of vision tasks, visual prompting large vision models are computationally intensive and resource-demanding due to their large parameter sizes and the complexity of processing visual prompts. In light of this view, the authors propose the Efficient Painter model, leveraging a novel context-aggregated attention-based trident
block to alleviate cross-task gaps and reduce memory and computation overhead.

**Strengths:**

1 The authors propose the Efficient Painter model which is faster and more memory-inefficient than Painter [Ref1].
2 The authors present ideas clearly, making the paper easy to follow.

**Weaknesses:**

1 The CAA passes the feature from the previous head to the next head. This will make heads can't run in parallel leading to increasing running time. Could you prove the effectiveness of this design compared to running them in parallel? More discussion should be added.

2 In CAA, the authors use a smaller dimension for Q and K. Could authors apply the same idea to ViT of Painter and check whether it can get similar results with fewer parameters? This actually challenges the novelty/efficiency of this proposed method, if setting the dimensions of Q, K to a lower dimension to ViT of Painter can reach similar results, then the proposed CAA might be a insufficient design.

3 The specialized models in Table 5 are outdated. The authors should compare the proposed model with the newest best-specialized methods.

4 Could authors evaluate the proposed model with other datasets (for generality), not only the datasets in Painter [Ref1]?

5 Why are the results of Painter (Efficient-ViT-M5) all NA in Table 5? Table 5 is extremely incomplete in this case, it becomes really hard to evaluate the performance across different methods systematically. I understand that  the benchmark might be hard to build, however, when reporting related performance, the relative results should be reported for completeness.

[Ref1]: Images Speak in Images: A Generalist Painter for In-Context Visual Learning.

**Questions:**

Please see Weaknesses.

---

> ### Author Response · Authors · 2024-11-27
>
> Dear Reviewer fA3M,
>
> Thank you for your thorough review and insightful comments on our manuscript. We appreciate your time and effort in providing detailed feedback, which has been invaluable in helping us improve our work. We have addressed each of your concerns below.
>
> **Weakness W1:**
>
> Thank you for pointing out the potential concern regarding the sequential processing in our Context-Aggregated Attention (CAA) module. **Our comprehensive analysis, as presented in Table 7, demonstrates that even though Multi-Head Self-Attention (MHSA) processes data in parallel, our CAA achieves superior computational efficiency under identical embedding dimensions.** The efficiency gains are achieved through multiple innovative strategies:
>
> - **Quantitative Comparison:** Given an input of intermediate window features ($x_p$) with shape ($[B, ED, H_p, W_p] = [1, 1024, 14, 14]$) and an identical embedding dimension ($ED = 1024$), our benchmarks show:
>
>     - **CAA:** 242.50 MFLOPs
>     - **MHSA:** 822.08 MFLOPs (3.07× higher)
>
> - **Dimension Reduction:** We utilize $\frac{1}{4}$ of the query (Q) and key (K) dimensions compared to traditional MHSA, significantly reducing computation without sacrificing performance.
>
> - **Spatially Localized Convolutional Embeddings:** By replacing dense fully connected embeddings (linear layers) with spatially localized convolutional embeddings, we reduce computational demand, which is particularly beneficial for resource-constrained devices during the training stage.
>
> These combined strategies result in at least a 4-fold improvement in computational efficiency across multiple stages of attention computation. **Moreover, despite the sequential nature of the heads in CAA, the overall running time does not increase significantly due to these optimizations.**
>
>
> **Weakness W2:**
>
> You raise a valuable point regarding the potential of simply reducing the Q and K dimensions in the ViT of Painter. **However, as shown in Figures 1(a) and 1(b) of our paper, our experiments with smaller QKV embedding dimensions of 768 using ViT-Base did not yield higher throughput.** Moreover, our empirical data reinforces this observation: even when reducing the dimensions of Q and K in MHSA from 64 to 16, the computational cost remains significantly higher compared to our proposed CAA design. Specifically, MHSA with fewer Q, K dimensions $64\rightarrow16$ requires 513.80 MFLOPs and has 2.62 million parameters, whereas CAA operates with only 242.50 MFLOPs and 1.22 million parameters. Therefore, simply reducing the Q and K dimensions of ViT-Large does not lead to substantial benefits in efficiency.
>
> **Weakness W3:**
>
> Thank you for your observation regarding the models included in Table 5, but we think it is unnecessary to compare our model with the newest best-specialized models. According to PromptGIP [1], the focus of visual prompt-based models is not to achieve state-of-the-art performance on every specialized task but to ensure strong in-context visual learning capabilities. Therefore, our goal is to modify visual prompt-based LVMs like Painter that possess strong in-context visual learning abilities, aiming to achieve both overall efficiency and excellent in-context visual learning performance. In the upper part of Table 5, we included comparisons with specialized models to further demonstrate that our efficient Painter not only achieves overall efficiency but also maintains robust in-context visual learning capabilities.
>
> [1] Liu, Y., Chen, X., Ma, X., Wang, X., Zhou, J., Qiao, Y. &amp; Dong, C.. (2024). Unifying Image Processing as Visual Prompting Question Answering. Proceedings of the 41st International Conference on Machine Learning, in Proceedings of Machine Learning Research

---

> ### Author Response · Authors · 2024-11-27
>
> **Weakness W4:**
>
> We appreciate your suggestion to evaluate our model on additional datasets to demonstrate generality. **Our model, like Painter and PromptGIP, performs inference validation based on the trained context.** Incorporating new task datasets would require us to finetune the model, which is challenging to achieve within a short timeframe due to computational and resource constraints.
>
> In this study, we focused on the datasets used in Painter [Ref1] to maintain consistency and facilitate direct comparisons. **We agree that extending our evaluations to include a wider variety of datasets would further validate the robustness and versatility of our model, and we will consider this in future work.**
>
> **Weakness W5:**
>
> Thank you for highlighting the issue with Table 5. **The inclusion of "NaN/NaN" placeholders for Painter (Efficient-ViT-M5) in Table 5** was intended to indicate that directly applying certain optimization methods designed for single-task models to visual prompt-based models like Painter without careful consideration is not applicable as baseline networks for comparison in our study, also doesn't meet the objective of our paper: to structurally refine a visual prompt-based LVM, aiming to simultaneously achieve both high overall efficiency and robust in-context visual learning capabilities.
>
> Although these designs effectively enhance model efficiency in specific tasks, they cannot be directly applied to visual prompt-based LVMs. They are mainly designed for single-task scenarios and lack extensibility to visual prompt mechanisms. Even if we forcibly apply these compression methods to Painter, they significantly impact its in-context visual learning capabilities. We demonstrate this by adapting the Painter architecture from ViT-Large to ViT-Base and fine-tuning it on three low-level datasets for comparative analysis—an experiment not conducted in the original Painter study. As shown in Figures 1. (a) and (b), the ViT-Base configuration did not significantly improve throughput and markedly compromised the model’s in-context visual learning capabilities.
>
> We acknowledge that the presentation of our experimental results needs to be more clear. In the revised version of the paper, we updated Table 5 to change the “NAN” placeholders for Painter (Efficient-ViT-M5) to “N/A” and added an explanation about this line of "N/A" placeholders for Painter (Efficient-ViT-M5).
>
> Once again, we appreciate your valuable feedback.

---

### Official Review · Reviewer_f6ai · 2024-11-03

**Soundness:** 3
**Presentation:** 2
**Contribution:** 2
**Rating:** 5
**Confidence:** 5

**Summary:**

This paper aims to address the computational inefficiencies and slow speeds resulting from the large number of parameters and complex prompt processing in visual promoting large vision models. The paper proposes a context-aggregate attention based on the trident block and a cross-blocks feature union. Compared to previous models, it is 9× smaller in model size and runs 4.1× and 27× faster during training and inference, respectively.

**Strengths:**

+ From the experimental results, the architecture of this paper shows certain improvements in computational efficiency and speed.
+ The designed architecture can be applied to multiple tasks, including denoising and low-light enhancement tasks.

**Weaknesses:**

- There are grammar issues in the writing, such as in the related work on the second page, ‘Traditional approaches such as DeiT, and MobileViT  leverage knowledge distillation or architectural downsize fail to enhance
actual inference speed or throughput.’
- Lack of innovation. The core contribution of this article is a lightweight architecture, but many of the operations that affect parameter changes are existing methods, such as integrating existing DWConv or reducing the dimension of queries.
- Non-standard use of abbreviations in the writing.

**Questions:**

1. What is the innovation in this article? In addition to existing dimension reduction methods, what is the innovative dimension reduction strategy proposed in this article?
2. From the description in Section 4.2 on the fourth page and Figure 3, The CAA module primarily computes attention weights for different channel features and enhances the response of important features. It then accumulates and stacks important features from different channels. How are deep connections established between different channels? More prominently observed in the current design is feature aggregation rather than the establishment of correlations.
3. The mention of aligning channel attention mechanisms on lines 204-205 can reduce computational costs. The related reasons need to be further elaborated with specificity.
4. The article mentions that in the design of sub-modules, smaller dimensions are used to reduce the parameter count. This is indeed a simple and effective strategy, but further experiments and analysis are needed for the core innovative operations that can reduce the parameters.
5. Please avoid using abbreviations for the first occurrence whenever possible.
6. In lines 310-316 on page six, it is explained that aligning the channels of the sub-decoder can alleviate gradient explosions. Please provide specific reasons why this operation is believed to mitigate gradient explosions.
7. The alpha and beta variables used in section 4.4 are not clearly elucidated regarding the numerical selection strategy. Experimental analysis of the parameters chosen is lacking.

---

> ### Author Response · Authors · 2024-11-27
>
> Dear Reviewer f6ai,
>
> Thank you for your thorough review and valuable feedback on our manuscript. We have addressed each of your points below.
>
> **Weaknesses W1 & W3 and Question Q5:**
>
> Thank you for bringing these writing and grammar issues to our attention. We tried our best to solve this in our revised version manuscript.
>
> **Weakness W2 & Question Q1.1:**
>
> **Our main innovation lies in the novel adaptation and integration of existing efficiency techniques within visual prompt-based LVMs.** This allows us to design a lightweight architecture that balances overall efficiency with the preservation of in-context visual learning capabilities, which traditional methods for single-task models do not address. Our initial experiments showed that simply adopting a lighter architecture—by replacing the ViT-large encoder with ViT-base or integrating a pretrained Efficient-ViT model—did not significantly improve throughput and notably degraded the model's in-context visual learning abilities, even causing severe instability like gradient explosion.
>
> By ensuring compatibility with multi-task, prompt-based settings, we enhance model efficiency without sacrificing visual contextual learning performance.
>
> **Question Q1.2:**
>
> **The dimensionality reduction methods** presented in this paper encompass not only traditional spatial reduction but also involve substituting the dense fully connected embeddings of traditional MHSA with spatially localized convolutional embeddings in our CAA module. This substitution effectively replaces the dense computations associated with fully connected embeddings. Assuming that both our CAA and MHSA use the same embedding dimension (ED = 1024) and the same number of heads, and under the same input of intermediate window features ($x_p$) with a shape of $[B, ED, H_p, W_p] = [1, 1024, 14, 14]$, our benchmark measurements demonstrate that CAA contains 242.50 MFLOPs, whereas MHSA contains approximately 822.08 MFLOPs—about 3.07 times higher than our CAA. Considering the actual implementation, our CAA further employs a smaller embedding dimension ED, which is consistent with the computational efficiency and speedup estimations theoretically calculated in Table 7.
>
> Additionally, **our CAA progressively reduces the computational complexity of the attention map using a channel-wise partition strategy** that segregates channel-independent contextual embedding information across different heads and employs depthwise convolutions to aggregate information from various channels.
>
> Lastly, owing to our CBFU's capability to capture global features, we can use fewer Trident Blocks to construct our primary backbone.
>
> By focusing on this trade-off between overall efficiency and in-context visual learning capabilities, we contribute a new perspective on enhancing the overall efficiency of visual prompt-based LVMs without sacrificing their contextual learning performance.
>
> **Question Q2:**
>
> Thank you for this insightful observation.
>
> **The CAA module is designed to establish deep connections between different channels through the attention weight calculation process, which goes beyond simple aggregation by modeling inter-channel relationships and correlations.** The deep connections are established through the following key mechanisms:
>
> **Iterative Channel Interaction Process:** As shown in Equation $F'j = F_j + \Phi(\tilde{F}{j-1})$, the CAA module element-wisely adds the different channel-wise split features $F_j$ to the convolutionally transformed channel features from the previous iteration, $\Phi(\tilde{F}_{j-1})$. This achieves efficient channel-wise connections of features from multiple different channels (excluding $F_0$) in the subsequent attention map calculation.
>
> **Multiple-channel Fusion in the Trident Block:** Within the Trident Block, we incorporate a multiple-channel fusion module that merges features from different channels from 2 merged contextual information. This fusion is not just a simple stacking of features but involves combining them in a way that captures the correlations and dependencies between channels. By doing so, the model learns to represent complex relationships among features.
>
> Our CAA module effectively captures both local and global inter-channel relationships, leading to a more expressive and robust feature representation.

---

> ### Author Response · Authors · 2024-11-27
>
> **Question Q3:**
>
> Thank you for pointing this out. The rationale behind this alignment is partly to make full use of limited computational resources. As shown in our visualization results (Fig. 4), MHSA without enough channel splits lacks sufficient contextual representation capability, while an appropriate increase in channel splits can enhance representation but may reduce efficiency.
>
> Meanwhile, aligning the number of channels with the number of attention heads allows us to effectively reduce the dimensionality of QKV embedding matrices. This adjustment allows us to decrease redundancy. It also minimizes redundant computations in the self-attention mechanism. Each head focuses on a distinct subset of features, which reduces the overall parameter count and FLOPs (floating point operations) required during both training and inference. As a result of these optimizations, we have demonstrated a 4.1× increase in training speed and a 27× increase in inference speed compared to traditional MHSA-based visual LVM.
>
> **By aligning channel attention mechanisms, we reduce computational costs by partitioning input feature channels to match individual attention heads, thereby reducing redundant computations within each head and lowering overall computational complexity.**
>
> **Question Q4:**
>
> Indeed, while certain submodules could be further analyzed to potentially reduce their sizes and decrease the number of parameters, the primary goal of our paper is to achieve overall efficiency while ensuring strong in-context visual learning capabilities. The current results already prove that our submodule design carefully balances these two objectives through key architectural choices.
>
> Moreover, conducting further ablation experiments requires retraining the model; however, the limited training time only allows us to verify that the experimental results meet all efficiency criteria, without sufficiently demonstrating the model's strong in-context visual learning capabilities. Due to resource limitations, we have provided an ablation study of our best model in Section 5.4, Table 6. We consider more detailed research and further optimization as objectives for future work.
>
> **Question Q6:**
>
> Thank you for this important question. Channel alignment is employed to preserve the original structure of the Painter decoder; on the other hand, by maintaining consistent feature dimensions across 3 major processing stages, it enables the use of more BatchNorm during the final stage of feature union. This reduces the variance in gradient magnitudes, mitigating the issue of gradient explosion, and facilitates improved gradient propagation, thereby enhancing the flow of information within the network's hierarchy.
>
> **Question Q7:**
>
> Thank you for pointing out the lack of clarity regarding the numerical selection strategy for the $\alpha$ and $\beta$ variables in Section 4.4. We appreciate the opportunity to elaborate on this aspect of our work.
> The parameter $\alpha$ controls the balance between the MS-SSIM loss and the L1-norm loss, while $\beta$ regulates the trade-off between the combined loss $\hat{L}$ and the context-aware feature alignment loss. We set $\alpha$ to 0.84, adopting this value from reference [1] because it effectively balances perceptual similarity and pixel-level accuracy as shown in prior studies. For $\beta$, we empirically set it to 0.5 to achieve a balanced influence between the combined loss and the contextual guidance from the ground-truth-based MS-SSIM loss, ensuring neither term dominates the training process.
>
> [1] H. Zhao, O. Gallo, I. Frosio, and J. Kautz, "Loss Functions for Image Restoration With Neural Networks," IEEE Transactions on Computational Imaging, vol. 3, no. 1, pp. 47-57, March 2017.
>
> We have revised Section 5.1 Implementation Details to explain the selected parameter values of our context-aware reconstruction loss function.
>
> Once again, we appreciate your valuable feedback.

---

### Official Review · Reviewer_XvQm · 2024-11-03

**Soundness:** 2
**Presentation:** 1
**Contribution:** 2
**Rating:** 3
**Confidence:** 3

**Summary:**

The authors propose an efficient large vision model (LVM) called Efficient Painter, designed to reduce memory and computational demands for visual tasks. Key contributions include a context-aggregated attention (CAA) mechanism that allows dynamic context processing with lower resource usage and a cross-block feature union (CBFU) module that improves multi-level global context processing, effectively speeding up training. The authors report significant reductions in memory consumption and substantial speed improvements during inference.

**Strengths:**

If understood correctly, the Efficient Painter model demonstrates substantial reductions in computational load and memory usage compared to its baselines. There appears to be a meaningful architectural novelty, especially in the design of the CAA and CBFU components, which seem well thought out and targeted toward efficiency.

**Weaknesses:**

The paper’s writing quality is quite limited, making it difficult to follow and understand key ideas, especially in the introduction, motivation, and experimental sections. The paper is filled with confusing sentences, which need substantial improvement for clarity and readability.


Moreover, the paper appears unfinished, with placeholders such as "NaN" in the result tables, which is especially noticeable in Table 5. The experimental results are presented in a messy and confusing manner, further detracting from the paper's overall quality.

**Questions:**

In its current state, I don’t believe this paper meets the standards for publication and would require significant revisions to be considered.

---

> ### Author Response · Authors · 2024-11-27
>
> Dear Reviewer XvQm,
>
> First, we sincerely apologize for any ambiguities and shortcomings in our paper, and we truly appreciate your thorough review and insightful feedback. We would like to take this opportunity to clarify the unclear parts of our paper.
>
> In the **Introduction**, we begin by introducing the concept of visual prompt-based Large Vision Models (LVMs) and highlight the distinctions between these models and Vision Language Models (VLMs). We then outline the challenges faced by visual prompt-based LVMs in performing in-context visual learning. Next, we discuss the limitations of some existing model compression designs. To address these challenges, we propose a design that strikes a balance between overall efficiency and in-context visual learning capabilities.
>
> The **motivation** of our paper is to ensure that the model retains robust in-context visual learning capabilities while improving overall efficiency. This rationale underpins our **experimental section**, which assesses not only computational and memory efficiency but also the model’s multi-task in-context learning(ICL) performance.
>
> **The inclusion of "NaN/NaN" placeholders for Painter (Efficient-ViT-M5) in Table 5** was intended to indicate that directly applying certain optimization methods designed for single-task models to visual prompt-based models like Painter without careful consideration is not applicable as baseline networks for comparison in our study, also doesn't meet the objective of our paper: to structurally refine a visual prompt-based LVM, aiming to simultaneously achieve both high overall efficiency and robust in-context visual learning capabilities.
>
> Although these designs effectively enhance model efficiency in specific tasks, they cannot be directly applied to visual prompt-based LVMs. They are mainly designed for single-task scenarios and lack extensibility to visual prompt mechanisms. Even if we forcibly apply these compression methods to Painter, they significantly impact its in-context visual learning capabilities. We demonstrate this by adapting the Painter architecture from ViT-Large to ViT-Base and fine-tuning it on three low-level datasets for comparative analysis—an experiment not conducted in the original Painter study. As shown in Figures 1. (a) and 1. (b), the ViT-Base configuration did not significantly improve throughput and markedly compromised the model’s ICL capabilities.
>
> We acknowledge that the presentation of our experimental results needs to be more clear. In the revised version of the paper, we updated Table 5 to change the “NAN” placeholders for Painter (Efficient-ViT-M5) to “N/A” and added an explanation about this line of "N/A" placeholders for Painter (Efficient-ViT-M5).
>
> Please let us know if you have any further questions or suggestions. Thank you once again for your valuable feedback.

---

### Author Response · Authors · 2024-11-27
**Global Response about paper revision details.**

In our revised manuscript, we have made several improvements to enhance clarity and better articulate the motivations and challenges of our work:

1. **Introduction – Second Paragraph:**
We have corrected grammatical issues to improve readability and have provided a more precise exposition of previous works and their limitations. Specifically, we have clarified that prior approaches not only fail to achieve overall efficiency but also overlook their applicability to visual prompt-based Large Vision Models (LVMs). This revision ensures a clearer understanding of the existing gaps that our research aims to address.

2. **Introduction – Third Paragraph:**
To highlight the primary challenge of our study more prominently, we underscore the delicate balance we aim to achieve between overall model efficiency (computation and memory efficiency) and the requisite in-context visual learning capacity for visual prompt-based models at the beginning of the paragraph. By tackling this challenge, we make our research objectives more immediately apparent to the reader.

3. **Section 3 – Motivation Part:**
We have added a new paragraph to the motivation section to delineate the process through which we identified the core problem. This addition clarifies the problem's significance and relevance and introduces our design stemming from this challenge.

4. **Section 5.1 - Implementation Details:**
We have revised this part to give more details about our selected parameter values in

5. **Abbreviation Clarifications:**
We have revised Section 5.1 Implementation Details to explain the selected parameter values of our context-aware reconstruction loss function.

We believe these revisions significantly enhance the manuscript by providing a clearer narrative and stronger justification for our research.

---

### Author Response · Authors · 2024-12-03
**Awaiting Feedback on Rebuttal Submission**

Dear Reviewers,

I hope this message finds you well. We eagerly await your feedback on our rebuttal and would greatly appreciate any additional comments or follow-up questions you might have regarding our work. Your insights will be invaluable for improving our research.

Best,
Authors

---

### Meta-Review · Area_Chair_gMpN · 2024-12-15

**Metareview:**

This paper introduces Efficient Painter, a large vision model designed to address computational inefficiencies in visual tasks. The key innovations include: (i) Context-Aggregated Attention: A mechanism for dynamic context processing that reduces memory and computational demands. (ii) Cross-Block Feature Union: A module for enhanced multi-level global context processing, improving efficiency during training and inference. The Efficient Painter model achieves significant reductions in computational load and memory usage compared to its baselines. However, several concerns remain. The paper’s writing quality is limited, making it challenging to follow, and the description of its main contribution, the CAA module, remains unclear. Considering the reviewers' feedback, the AC has decided to reject this paper.

**Additional Comments On Reviewer Discussion:**

Reviewer XvQm highlighted significant issues with the paper's writing quality, making it difficult to follow and understand the key ideas. The updated version did not fully address these concerns.

Reviewer f6ai raised concerns about the lack of innovation and poor writing. The rebuttal did not provide sufficient evidence to resolve these issues.

Reviewer JQqw initially gave a positive score of 6 but noted concerns about the unclear description of the proposed CAA and the potential complexity of the method. These concerns remain unresolved despite the rebuttal, indicating the need for further manuscript revision.

---

### Decision · Program_Chairs · 2025-01-22

Reject